# Effects of health literacy, screening, and participant choice on action plans for reducing unhealthy snacking in Australia: A randomised controlled trial

Julie Ayre[1], Erin Cvejic[1,2], Carissa Bonner[1], Robin M. Turner[3], Stephen D. Walter[4], Kirsten J. McCaffery[1]*

1 Sydney Health Literacy Lab, Faculty of Medicine and Health, School of Public Health, University of Sydney, Sydney, New South Wales, Australia, 2 Faculty of Medicine and Health, School of Public Health, University of Sydney, Sydney, New South Wales, Australia, 3 Centre for Biostatistics, Division of Health Sciences, University of Otago, Dunedin, New Zealand, 4 Department of Health Research Methods, Evidence, and Impact, Faculty of Health Sciences, McMaster University, Hamilton, Ontario, Canada

* Kirsten.mccaffery@sydney.edu.au

**Data Availability Statement:** De-identified data are available in S1 Data.

## Abstract

### Background

Low health literacy is associated with poorer health outcomes. A key strategy to address health literacy is a universal precautions approach, which recommends using health-literate design for all health interventions, not just those targeting people with low health literacy. This approach has advantages: Health literacy assessment and tailoring are not required. However, action plans may be more effective when tailored by health literacy. This study evaluated the impact of health literacy and action plan type on unhealthy snacking for people who have high BMI or type 2 diabetes (Aim 1) and the most effective method of action plan allocation (Aim 2).

### Methods and findings

We performed a 2-stage randomised controlled trial in Australia between 14 February and 6 June 2019. In total, 1,769 participants (mean age: 49.8 years [SD = 11.7]; 56.1% female [$n$ = 992]; mean BMI: 32.9 kg/m² [SD = 8.7]; 29.6% self-reported type 2 diabetes [$n$ = 523]) were randomised to 1 of 3 allocation methods (random, health literacy screening, or participant selection) and 1 of 2 action plans to reduce unhealthy snacking (standard versus literacy-sensitive). Regression analysis evaluated the impact of health literacy (Newest Vital Sign [NVS]), allocation method, and action plan on reduction in self-reported serves of unhealthy snacks (primary outcome) at 4-week follow-up. Secondary outcomes were perceived extent of unhealthy snacking, difficulty using the plans, habit strength, and action control. Analyses controlled for age, level of education, language spoken at home, diabetes status, baseline habit strength, and baseline self-reported serves of unhealthy snacks. Average NVS score was 3.6 out of 6 (SD = 2.0). Participants reported consuming 25.0 serves of snacks on average per week at baseline (SD = 28.0). Regarding Aim 1, 398 participants in the random allocation arm completed follow-up (67.7%). On average, people scoring 1 SD

**Funding:** This research is supported by an Australian Government Research Training Program (RTP) Scholarship awarded to JA (https://www.education.gov.au/research-training-program); and Diabetes Australia, grant number G199678 (https://www.diabetesaustralia.com.au/diabetes-australia-research-program), awarded to CB, JA, KM and RT. The funders had no role in study design, data collection and analysis, decision to publish, or preparation of the manuscript.

**Competing interests:** The authors have declared that no competing interests exist.

**Abbreviations:** CSIRO, Commonwealth Scientific and Industrial Research Organisation; NVS, Newest Vital Sign; RCT, randomised controlled trial.

below the mean for health literacy consumed 10.0 fewer serves per week using the literacy-sensitive action plan compared to the standard action plan (95% CI: 0.05 to 19.5; $p = 0.039$), whereas those scoring 1 SD above the mean consumed 3.0 fewer serves using the standard action plan compared to the literacy-sensitive action plan (95% CI: −6.3 to 12.2; $p = 0.529$), although this difference did not reach statistical significance. In addition, we observed a non-significant action plan × health literacy (NVS) interaction ($b = -3.25$; 95% CI: −6.55 to 0.05; $p = 0.054$). Regarding Aim 2, 1,177 participants across the 3 allocation method arms completed follow-up (66.5%). There was no effect of allocation method on reduction of unhealthy snacking, including no effect of health literacy screening compared to participant selection ($b = 1.79$; 95% CI: −0.16 to 3.73; $p = 0.067$). Key limitations include low–moderate retention, use of a single-occasion self-reported primary outcome, and reporting of a number of extreme, yet plausible, snacking scores, which rendered interpretation more challenging. Adverse events were not assessed.

## Conclusions

In our study we observed nominal improvements in effectiveness of action plans tailored to health literacy; however, these improvements did not reach statistical significance, and the costs associated with such strategies compared with universal precautions need further investigation. This study highlights the importance of considering differential effects of health literacy on intervention effectiveness.

## Trial registration

Australia and New Zealand Clinical Trial Registry ACTRN12618001409268.

## Author summary

### Why was this study done?

- Health literacy is increasingly recognised as an important contributor to health outcomes.

- Most health literacy interventions aim to make information easier to understand, but few behaviour change strategies address health literacy issues explicitly, and few studies have investigated the effect of an individual's health literacy on behaviour change strategies.

- This study aimed to evaluate (1) the effects of health literacy and literacy-sensitive design on the effectiveness of action plans to reduce unhealthy snacking and (2) the most effective method of allocating an action plan.

### What did the researchers do and find?

- In total, 1,769 participants were randomised to 1 of 3 allocation methods (random, health literacy screening, participant selection) and 1 of 2 action plans to reduce unhealthy snacking (standard versus literacy-sensitive), with follow-up after 4 weeks.

- The literacy-sensitive action plan may be more effective for participants with low/moderate health literacy. We observed that the standard action plan may reduce snack consumption for those with high health literacy; however, the interaction between health literacy and action plan did not reach statistical significance.

- We did not find evidence suggesting that allocation method had a significant effect on reported snacking behaviour.

### What do these findings mean?

- There may be a benefit to tailoring action plans to individual health literacy, particularly for those with lower health literacy.

- However, this benefit did not reach statistical significance and should be weighed against the cost of tailoring compared to providing everyone with the same (literacy-sensitive) intervention (universal precautions approach).

- Future research should continue to investigate how health literacy and literacy-sensitive designs impact on intervention effectiveness.

## Introduction

Health literacy describes the cognitive and social skills that an individual uses to access, understand, and act on health information [1]. Low health literacy is increasingly recognised as a prevalent and notable contributor to health inequality around the world, including in Australia, Europe, and the United States [2–4]. It is associated with increased hospitalisation, mortality, and prevalence of chronic disease and risk factors for health conditions [5].

The main approach to addressing low health literacy in the design of health information is to reduce the cognitive demand of all health resources [6], which is most easily achieved by simplifying health information so that it is easier to understand for everyone [5–7]. Often this involves strategies such as using simple language and images, and breaking information down into smaller, more manageable amounts [7]. However, little emphasis is placed on reducing the cognitive demand of behaviour change strategies that promote actions such as planning, self-monitoring, and problem solving [8,9]. This represents an important research gap, particularly in the context of chronic conditions (including diabetes), as strategies that promote action play a key role in self-management behaviours such as healthy eating and physical activity [10–14].

In terms of implementation, this approach also recommends that all consumers can benefit from a health-literate intervention design [7]. This 'universal precautions' approach is recognised as a key strategy to addressing health literacy at an international level (e.g., by the World Health Organization [4]) and at a national level (e.g., the Department of Health and Human Services in the US [15] and the Australian Commission on Safety and Quality in Healthcare in Australia [16]). This approach has been shown to increase comprehension of health information and knowledge uptake for people with lower health literacy [5,17–19]. It also has practical advantages, because there is no need to assess a person's health literacy, nor to provide and distribute tailored interventions. However, it is important to consider that in applying a universal precautions approach to behavioural interventions, we implicitly assume that all consumers

will benefit from strategies to promote action that are less cognitively demanding. This may not be the case; at least in some cases, cognitive effort has shown to increase engagement in these kinds of tasks [20,21]. To date, few studies have evaluated whether applying health literacy principles to strategies promoting action has different effects for people with low and high health literacy [5,17,19,22].

Our recent randomised controlled trial (RCT) addressed both of these gaps in the literature [23]. We evaluated the impact of applying health literacy principles to an online action plan to reduce self-reported unhealthy snacking over a 4-week period. The 'literacy-sensitive' version of the action plan incorporated health literacy strategies such as using simple language and images, and guided participants step-by-step to create an effective plan. This involved identifying a 'snack moment' and a solution (a behaviour to replace snacking) from predetermined lists. This was compared to a 'standard' action plan that asked participants to formulate their own free-text plans, affording greater potential for personalisation. The results showed that the effectiveness of each plan depended on the participant's health literacy: people with lower health literacy reported consuming fewer unhealthy snacks at follow-up when they had used the literacy-sensitive action plan, whereas people with higher health literacy reported consuming fewer unhealthy snacks using the 'standard' action plan [23].

These findings contrast with the assumptions of the universal precautions approach and instead suggest that strategies to promote action may need to be tailored to a person's health literacy level. However, this is a novel area of research that warrants investigation in additional, varied samples, especially in light of the current reproducibility crisis across health and psychology [24]. The current study aimed to build on the previous study in 2 ways: (1) to evaluate the impact of health literacy and type of action plan ('literacy-sensitive' or 'standard') on unhealthy snacking in a sample for whom reduced snacking would be of clinical benefit and (2) to evaluate the effect of method of allocation to either a literacy-sensitive or standard action plan on overall effectiveness of the intervention.

For people with type 2 diabetes and/or and overweight or obese body mass index (BMI), reducing unhealthy snacking is of clinical benefit and may assist with weight management [25–27]. Weight management has been shown to improve insulin sensitivity, fasting blood glucose, blood pressure, and lipid levels, and to help reduce the need for diabetes medications [28–30].

To achieve our second aim, we evaluated 3 alternative methods of allocating participants to an action plan: random allocation, screening for low health literacy, or allowing the participant to choose. Screening for health literacy is an obvious option given the results of our previous study [23]. However, health literacy assessment may also increase participant burden and reduce engagement with the intervention. Another option would be to allow the participant to select their preferred action plan. Although it is unlikely that participants would always select an action plan that matched their health literacy level, this approach has other potential compensatory advantages; for example, choice has been shown to increase participant satisfaction with an intervention, and to increase participant motivation, effort, and task performance [31–33]. The case for this argument is supported by findings from RCTs that indicate interventions are more effective for participants who receive their preferred intervention [34].

In view of this literature, we hypothesised that [35] (1) a literacy-sensitive action plan would be more effective at reducing unhealthy snacking for participants with lower health literacy, whereas a standard action plan would be more effective for participants with higher health literacy, and (2) the intervention would be more effective at reducing unhealthy snacking for participants who were allocated an action plan using the health literacy screening tool, compared with those asked to select their preferred action plan, and both of these allocation methods would be more effective than random allocation to an action plan.

## Methods

### Ethics statement

This study was approved by the University of Sydney Human Research Ethics Committee [2018/793]. As specified in the participant information sheet, participants indicated consent by completing the online survey.

### Study design

This is a 2-stage online RCT that compared the effects of health literacy, action plan type, and method of allocation to an action plan (random, screened, and choice) on self-reported unhealthy snacking over 1 month (Fig 1). A third research question involving Arm A1 in Fig 1 (effect of assessment of preference) and preference and selection effects, as outlined in the protocol papers [35,36] (see also Table A in S1 Text), will be addressed elsewhere. The study was prospectively registered with ANZCTR (http://www.anzctr.org.au) (ACTRN12618001409268). The protocol and CONSORT and TIDieR checklists are shown in S1 Text (Tables A, B, and C, respectively).

### Participants and recruitment

Participants with an Australian internet protocol (IP) address were recruited through an online market research company, between 14 February 2019 and 6 June 2019, for a total of 2,584 participants (Fig 1). Inclusion criteria were self-reported diagnosis of type 2 diabetes and/or self-reported height and weight corresponding to an overweight or obese BMI (i.e., BMI ≥ 25 kg/m$^2$). To maximise the proportion of the sample with type 2 diabetes, people with diabetes were actively recruited first, after which the recruitment strategy was broadened to

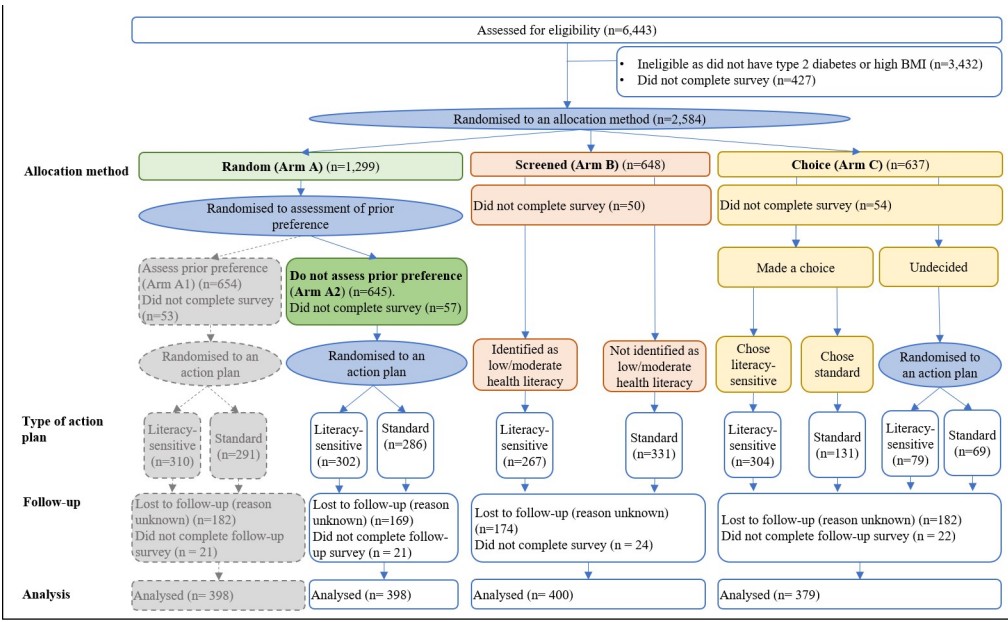

**Fig 1. Participant flowchart.** Aim 1 was assessed using Arm A2 only. Aim 2 was assessed using Arms A2, B and C. Details and results relating to Arm A1 are not the focus of the research questions in this paper and are not be reported here. Numbers indicated under 'type of action plan' sum to the total baseline sample who created an action plan and were included in the baseline sample ($n$ = 1,769). Numbers indicated under 'analysis' sum to number of participants who completed follow-up ($n$ = 1,177).

include those with overweight or obese BMI. Exploiting the fact that the education levels of potential participants were available from the market research company, we recruited at least 70% of participants with less than university education to ensure a diverse range of health literacy levels. Participants were not eligible if they did not speak English. No participants were excluded on the basis of their snacking behaviour (e.g., participants with extreme, yet plausible, snacking scores were not excluded). Participants received an information statement about the study that indicated that they would be giving informed consent by virtue of their completing the online survey. Because the survey was released simultaneously to a large number of potential participants, a number of them began but did not complete the survey before the total sample size quota was filled (Fig 1). There were small differences in the number of participants randomised to each allocation method arm (Fig 1). This was likely due to a combination of factors, e.g., total quota filled after a participant was randomised, a small number of participants who completed the survey multiple times, or participants who did not provide a valid email address for the follow-up survey. Baseline and follow-up surveys are available in S1 Surveys.

## Participant allocation

After completing baseline measures, participants were randomised to 1 of 3 allocation methods [35]. The allocation method then determined how they would be assigned an action plan, either literacy-sensitive (designed for people with lower health literacy) or standard.

## Allocation method

**Random (Arm A2).** Participants were randomised to the standard or literacy-sensitive action plan.

**Screened (Arm B).** Participants with inadequate health literacy (low/moderate according to the Newest Vital Sign [NVS]) were assigned the literacy-sensitive action plan, and the remaining participants were assigned the standard action plan.

**Choice (Arm C).** Participants were given a brief description of each action plan and then could choose which one they would like to use. Those who indicated they were 'unsure' were given longer descriptions of the 2 plans before being asked again to make a selection. Participants who indicated they were still unsure were randomised to an action plan.

## Action plan interventions

**Literacy-sensitive action plan: Smart Snacking 101 (basic).** This intervention was designed for people with lower health literacy. The intervention guides participants through the process of developing a plan (also shown in Fig 2 in the protocol paper [35]). Participants were asked to select from a list 3 situations in which they felt that they often ate unhealthy snacks or ate too much, and then to select the situation they would be happiest to change. Participants were then asked to select 1 option from a list of possible behaviours ('plans') they could take in this challenging situation. After making this selection, participants were asked to imagine how it might feel to enact the plan, and to rate how difficult they thought this would be on a scale from 1 (very easy) to 10 (very hard). Participants who indicated a difficulty level greater than or equal to 7 were advised to select a different plan from the list, in order to reduce their perceived difficulty of enacting the plan. Health literacy strategies such as using simple language, white space, and illustrative images were incorporated into the design [7].

**Standard action plan: Smart Snacking Pro (advanced).** Participants were asked to formulate a plan to reduce their unhealthy snacking in as much detail as possible, paying specific attention to the situations in which they would implement their plan. Participants entered their selected situations and plan into text boxes.

**A: All observations**

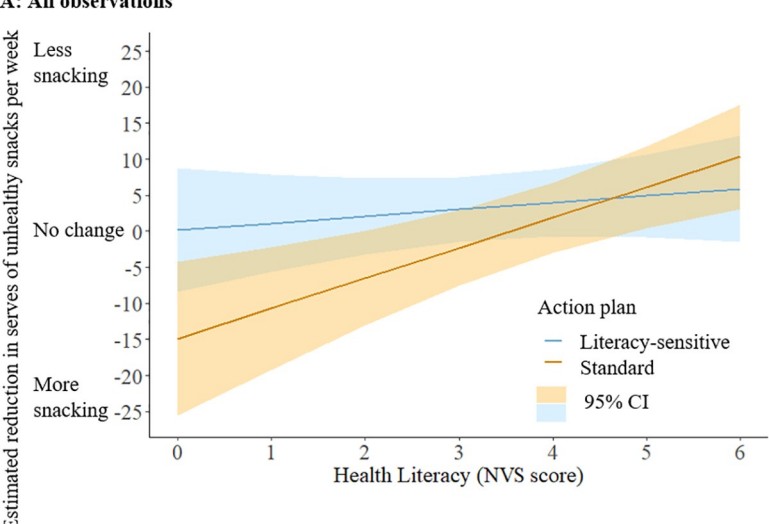

**B: Outlier and influential observations removed**

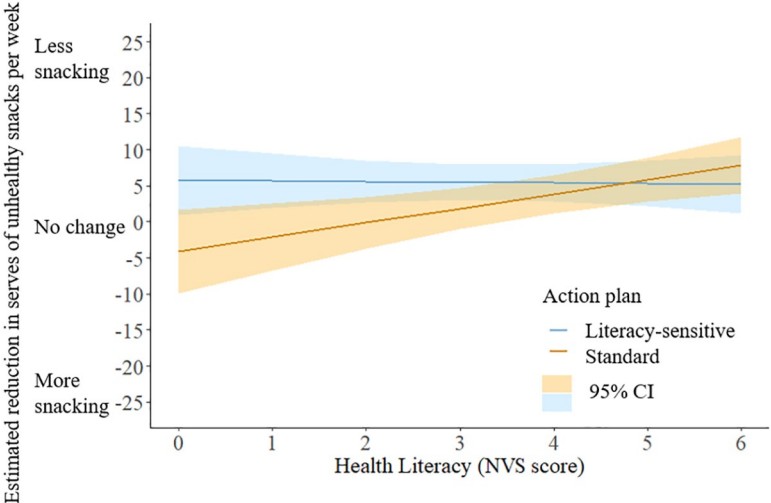

**Fig 2. Estimated reduction in serves of unhealthy snacks per week by action plan and health literacy score (randomised arm).** (A) All observations. (B) Outlier and influential observations removed. Estimates are for covariates (age, language spoken at home, education, diabetes status, baseline habit strength, and baseline snacking score) controlled at mean value for the randomised arm. Bands indicate 95% confidence intervals. NVS, Newest Vital Sign.

Participants were presented with a copy of their completed action plan and instructed to write it down, take a screenshot, or download it as a digital copy. They were also asked to confirm that they had a copy [35].

## Baseline and follow-up surveys

At baseline, participants provided their demographic information, and completed questions assessing snacking behaviour, habit strength, and intentions to reduce unhealthy snacking. Participants were then randomised to an allocation method and created a plan. Intention to reduce unhealthy snacking was measured a second time, immediately after creating the plan. During the first, second, and third weeks, participants were emailed a reminder of their plan.

After 4 weeks participants were asked to complete a follow-up survey assessing snacking behaviour, habit strength, intention, and action control.

## Measures

**Demographic measures.**   Participants were asked to complete questions about their age, employment status, highest level of educational attainment, and, if they reported having diabetes, years since diagnosis and whether they used insulin [35].

**Health literacy.**   Health literacy was measured using the NVS [37] (6-item measure of functional health literacy; Cronbach's α = 0.77) and a single-item literacy screener [38]. The NVS is a reliable instrument for assessing health literacy that has been validated through comparison to the Test of Functional Health Literacy in Adults (TOFHLA), a gold-standard measure of functional health literacy [38]. NVS scores of 0 and 1 indicate a high likelihood of limited healthy literacy, scores of 2 and 3 indicate the possibility of limited health literacy, and scores of 4–6 indicate adequate health literacy. In this paper we refer to these categories as 'low', 'moderate', and 'high', respectively.

**Primary outcome.**   Snacking scores were measured using a validated diet score instrument developed by the Commonwealth Scientific and Industrial Research Organisation (CSIRO) [39]. Items were drawn from the 'discretionary foods' category, which the Australian Guide to Healthy Eating defines as foods 'not considered necessary for a healthy diet' [40]. The guidelines define 1 serve of discretionary foods as an amount that contains approximately 600 kJ. Alcohol was excluded from the assessment in this study as the focus was on 'snacks'. Participants were asked to reflect on their snacking behaviour in the past month for 9 categories of discretionary foods. Participants selected a timeframe (day, week, month, or 'I don't eat this') and indicated the number of serves for this timeframe (up to a maximum 10 serves). The average weekly serves of unhealthy snacks was calculated from these scores. The primary outcome was reduction in unhealthy snacking between baseline and follow-up (i.e., a positive number indicates a reduction in the snacking score).

**Secondary outcomes.**   Difficulty using the planning tool was assessed using a 5-point scale (from 1 = not at all hard, to 5 = extremely hard). This was assessed immediately following the intervention. This instrument was analysed as 3 categories: 'not at all hard', 'a little hard', and 'somewhat hard' to 'extremely hard'. The remaining secondary outcomes were assessed at 4-week follow-up. Perceived extents of healthy and unhealthy snacking in the previous week were each assessed using a single-item 7-point Likert scale (from 1 = not at all healthy/unhealthy, to 7 = very much healthy/unhealthy). Intention to reduce unhealthy snacking was assessed using 3 items [41,42], habit strength was assessed using a 12-item self-report habit strength index [43], and action control (strategies to self-regulate unhealthy snacking [44]) was assessed using a 6-item measure [44]. The items for these last 3 outcomes were assessed using a 7-point Likert scale (strongly disagree to strongly agree). Difficulty following the plan (Likert scale ranging from 0 = very easy, to 10 = very hard) and action plan preference were also assessed at 4-week follow-up but will be addressed elsewhere.

## Sample size

Estimates of expected effect size and retention were based on previous work exploring the effects of action plans on unhealthy snacking [23]. A sample size estimate of 500 in each allocation arm was based on requiring 80% power using a 5% significance level to detect a small main effect (Cohen's *f* ratio effect size = 0.08, as in the previous study) in a univariate ANOVA comparing the 3 allocation method arms. This corresponds to a minimum pairwise difference between groups of approximately 0.18 standard deviations. We anticipated 15% attrition between baseline and follow-up, and therefore estimated that a total of 1,764 participants should be recruited at baseline into the random, choice, and screened arms to ensure an adequate sample size.

## Analysis

For each of the following regressions, unadjusted and adjusted models are presented. $p$-Values $< 0.05$ were considered statistically significant. Intention-to-treat analysis was conducted using worst-case (no change in snacking score) and best-case (average change in snacking score for that allocation method × action plan combination) scenarios for participants who did not complete follow-up. Baseline characteristics of completers and non-completers were compared to assess bias and generalisability. In addition, diagnostic statistics were used to identify influential observations for each regression model. These were based on standardised residuals, Cook's distance, leverage values, and standardised DFbeta coefficients. We conducted a sensitivity analysis by sequentially removing influential observations individually from the model to check the robustness of the model.

**Aim 1: Effects of health literacy and action plan.**    This analysis aimed to examine the effects of health literacy and randomisation to an action plan. Multiple linear regression examined change in snacking scores for participants in the randomised arm (Arm A2). Predictors included action plan, health literacy, and an action plan × health literacy interaction term. The model controlled for important correlates of health literacy (age, level of education, language spoken at home) [5], diabetes status, and baseline snacking score [35]. NVS scores were examined both continuously and categorically (i.e., inadequate versus adequate health literacy). Given the high number of participants reporting that the tool was 'not at all hard' to use, perceived difficulty using the plan was analysed using a chi-squared test.

**Aim 2: Effects of allocation method.**    The primary analysis used multiple linear regression to evaluate the change in snacking scores between the 3 arms (random, screened, and choice) whilst adjusting for any effect of diabetes status. Two orthogonal contrasts were used. The first compared outcomes in the random arm to those in the choice/screened arms. The second compared outcomes in the choice arm to those in the screened arm. The analysis was repeated on the secondary outcomes of perceived unhealthy snacking in the previous week, difficulty using the planning tool, action control, and habit strength. A sub-group analysis was conducted on participants with or without type 2 diabetes.

**Additional qualitative analysis of action plan content.**    Two researchers independently coded standard action plans in the randomised arm according to (1) the extent that the plans adhered to the instructions (i.e., provided at least 1 'situation' and 1 'plan') and (2) the extent that situations and plans included characteristics that would not have been available using the literacy-sensitive action plan. Coders were blind to the health literacy level of participants. Any disagreements were systematically resolved through discussion.

## Results

### Sample characteristics

Results refer to the 3 arms related to the research questions explored in this paper (random, screened, and choice). Descriptive statistics for the full sample are available in S2 Text (i.e., including the arm where participants' preferences for an action plan were assessed prior to random allocation to an action plan; Tables A and B in S2 Text).

There were 1,769 participants at baseline, of whom 523 (29.6%) reported a diagnosis of type 2 diabetes (Table 1). Across the 3 arms, participants were an average 49.8 years of age (SD = 11.7), 56.1% were female ($n = 992$), and average BMI was in the obese range (BMI $> 30$ kg/m$^2$) (mean = 3/2.9, SD = 8.7). On average, participants at baseline reported consuming 25.0 serves of snacks per week (SD = 28.0) (Table 2), and average intentions to reduce unhealthy snacking were above the midpoint on the scale, indicating moderately positive intentions

Table 1. Baseline participant characteristics by allocation method.

| Demographic variable | Random | | | | | | Choice | | | | | | Screened | | | | | |
|---|---|---|---|---|---|---|---|---|---|---|---|---|---|---|---|---|---|---|
| | Literacy-sensitive action plan | | Standard action plan | | Total | | Literacy-sensitive action plan | | Standard action plan | | Total | | Literacy-sensitive action plan | | Standard action plan | | Total | |
| | N or mean | Percent or SD | N or mean | Percent or SD | N or mean | Percent or SD | N or mean | Percent or SD | N or mean | Percent or SD | N or mean | Percent or SD | N or mean | Percent or SD | N or mean | Percent or SD | N or mean | Percent or SD |
| **Female** | 170 | 56.3 | 164 | 57.3 | 334 | 56.8 | 189 | 49.3 | 129 | 64.5 | 318 | 54.5 | 134 | 50.2 | 206 | 62.2 | 340 | 56.9 |
| **Speaks English at home** | 297 | 98.3 | 280 | 97.9 | 577 | 98.1 | 377 | 98.4 | 199 | 99.5 | 576 | 98.8 | 262 | 98.1 | 327 | 98.8 | 589 | 98.5 |
| **Education** | | | | | | | | | | | | | | | | | | |
| Less than high school education | 20 | 6.6 | 28 | 9.8 | 48 | 8.2 | 36 | 9.4 | 19 | 9.5 | 55 | 9.4 | 33 | 12.4 | 21 | 6.3 | 54 | 9.0 |
| High school graduate | 93 | 30.8 | 97 | 33.9 | 190 | 32.3 | 126 | 32.9 | 54 | 27.0 | 180 | 30.9 | 91 | 34.1 | 95 | 28.7 | 186 | 31.1 |
| Certificate | 101 | 33.4 | 101 | 35.3 | 202 | 34.4 | 129 | 33.7 | 67 | 33.5 | 196 | 33.6 | 74 | 27.7 | 118 | 35.6 | 192 | 32.1 |
| University education | 88 | 29.1 | 60 | 21.0 | 148 | 25.2 | 92 | 24.0 | 60 | 30.0 | 152 | 26.1 | 69 | 25.8 | 97 | 29.3 | 166 | 27.8 |
| **Health literacy (NVS)** | | | | | | | | | | | | | | | | | | |
| Low (score 0 or 1) | 60 | 19.9 | 50 | 17.5 | 110 | 18.7 | 76 | 19.8 | 40 | 20.0 | 116 | 19.9 | 126 | 47.2 | 0 | 0.0 | 126 | 21.1 |
| Moderate (score 2 or 3) | 74 | 24.5 | 64 | 22.4 | 138 | 23.5 | 89 | 23.2 | 49 | 24.5 | 138 | 23.7 | 141 | 52.8 | 0 | 0.0 | 141 | 23.6 |
| High (score 4–6) | 168 | 55.6 | 172 | 60.1 | 340 | 57.8 | 218 | 56.9 | 111 | 55.5 | 329 | 56.4 | 0 | 0.0 | 331 | 100.0 | 331 | 55.4 |
| **Self-reported BMI (kg/m²)** | | | | | | | | | | | | | | | | | | |
| Underweight (<18.5) | 1 | 0.3 | 0 | 0.0 | 1 | 0.2 | 4 | 1.0 | 0 | 0.0 | 4 | 0.7 | 0 | 0.0 | 1 | 0.3 | 1 | 0.2 |
| Normal weight (18.5–24.9) | 12 | 4.0 | 10 | 3.5 | 22 | 3.7 | 9 | 2.3 | 6 | 3.0 | 15 | 2.6 | 15 | 5.6 | 10 | 3.0 | 25 | 4.2 |
| Overweight (25.0–29.9) | 121 | 40.1 | 115 | 40.2 | 236 | 40.1 | 152 | 39.7 | 93 | 46.5 | 245 | 42.0 | 104 | 39.0 | 132 | 39.9 | 236 | 39.5 |
| Obese (≥30.0) | 168 | 55.6 | 161 | 56.3 | 329 | 56.0 | 218 | 56.9 | 101 | 50.5 | 319 | 54.7 | 148 | 55.4 | 188 | 56.8 | 336 | 56.2 |
| **Self-reported diagnosis of type 2 diabetes** | 88 | 29.1 | 81 | 28.3 | 169 | 28.7 | 112 | 29.2 | 60 | 30.0 | 172 | 29.5 | 82 | 30.7 | 100 | 30.2 | 182 | 30.4 |
| Self-reported use of insulin | 25 | 28.4 | 18 | 22.2 | 43 | 25.4 | 26 | 23.2 | 15 | 25.0 | 41 | 23.8 | 28 | 34.1 | 23 | 23.0 | 51 | 28.0 |
| Years since diagnosis (mean, SD) | 9.7 | 7.4 | 9.2 | 7.7 | 9.5 | 7.5 | 9.6 | 8.4 | 7.4 | 6.5 | 8.8 | 7.8 | 10.7 | 8.1 | 10.2 | 7.7 | 10.4 | 7.9 |
| **Total** | 302 | 51.4 | 286 | 48.6 | 588 | | 383 | 65.7 | 200 | 34.3 | 583 | | 267 | 44.6 | 331 | 55.4 | 598 | |

BMI, body mass index; NVS, Newest Vital Sign.

**Table 2. Outcome variables at baseline and follow-up by allocation method.**

| Outcome | Random | | | | | | Choice | | | | | | Screened | | | | | |
|---|---|---|---|---|---|---|---|---|---|---|---|---|---|---|---|---|---|---|
| | Literacy-sensitive action plan | | Standard action plan | | Total | | Literacy-sensitive action plan | | Standard action plan | | Total | | Literacy-sensitive action plan | | Standard action plan | | Total | |
| | Mean or N | SD or percent | Mean or N | SD or percent | Mean or N | SD or percent | Mean or N | SD or percent | Mean or N | SD or percent | Mean or N | SD or percent | Mean or N | SD or percent | Mean or N | SD or percent | Mean or N | SD or percent |
| **Snacking score (serves per week)** | | | | | | | | | | | | | | | | | | |
| Baseline | 27.4 | 33.7 | 27.3 | 41.9 | 27.4 | 37.9 | 24.0 | 24.9 | 26.8 | 37.3 | 24.9 | 29.7 | 25.9 | 41.9 | 24.7 | 26.4 | 25.3 | 34.2 |
| Follow-up | 24.4 | 39.3 | 23.2 | 43.5 | 23.9 | 41.2 | 24.9 | 32.1 | 24.8 | 47.0 | 24.9 | 37.7 | 22.2 | 26.8 | 19.2 | 21.1 | 20.6 | 23.9 |
| Difference | -2.5 | 31.4 | -2.1 | 47.2 | -2.3 | 39.4 | 0.8 | 23.6 | -0.5 | 26.3 | 0.3 | 24.5 | -1.3 | 26.8 | -4.6 | 22.8 | -3.0 | 24.8 |
| **Perceived extent of healthy snacking (1 [low] to 7 [high])** | | | | | | | | | | | | | | | | | | |
| Baseline | 3.7 | 1.9 | 3.7 | 2.0 | 3.7 | 2.0 | 3.6 | 1.9 | 3.9 | 1.9 | 3.7 | 1.9 | 4.2 | 2.0 | 3.9 | 1.9 | 4.0 | 2.0 |
| Follow-up | 3.8 | 1.9 | 3.8 | 2.0 | 3.8 | 1.9 | 3.8 | 2.0 | 4.2 | 2.1 | 3.9 | 2.0 | 4.0 | 1.9 | 3.9 | 1.8 | 4.0 | 1.9 |
| Difference | 0.0 | 1.8 | 0.0 | 2.0 | 0.0 | 1.9 | 0.2 | 1.7 | 0.3 | 1.7 | 0.2 | 1.7 | -0.1 | 1.9 | 0.1 | 1.6 | 0.0 | 1.8 |
| **Perceived extent of unhealthy snacking (1 [low] to 7 [high])** | | | | | | | | | | | | | | | | | | |
| Baseline | 3.6 | 2.0 | 3.3 | 2.1 | 3.5 | 2.0 | 3.4 | 1.9 | 3.4 | 2.1 | 3.4 | 2.0 | 3.0 | 1.9 | 3.3 | 2.0 | 3.2 | 2.0 |
| Follow-up | 3.5 | 2.0 | 3.2 | 2.1 | 3.4 | 2.0 | 3.3 | 1.9 | 3.3 | 2.1 | 3.3 | 2.0 | 3.3 | 2.0 | 3.3 | 2.0 | 3.3 | 2.0 |
| Difference | 0.0 | 1.7 | 0.0 | 1.9 | 0.0 | 1.8 | -0.1 | 1.7 | -0.1 | 1.7 | -0.1 | 1.7 | 0.2 | 2.0 | -0.1 | 1.7 | 0.0 | 1.8 |
| **Intention to reduce unhealthy snacking (1 [low] to 7 [high])** | | | | | | | | | | | | | | | | | | |
| Baseline | 5.4 | 1.3 | 5.4 | 1.3 | 5.4 | 1.3 | 5.2 | 1.2 | 5.4 | 1.4 | 5.3 | 1.3 | 5.2 | 1.4 | 5.5 | 1.3 | 5.4 | 1.3 |
| Follow-up | 5.2 | 1.3 | 5.1 | 1.3 | 5.1 | 1.3 | 5.1 | 1.2 | 5.3 | 1.3 | 5.2 | 1.3 | 5.0 | 1.4 | 5.4 | 1.2 | 5.2 | 1.3 |
| Difference | -0.2 | 1.2 | -0.1 | 1.3 | -0.2 | 1.3 | 0.0 | 1.1 | -0.1 | 1.2 | 0.0 | 1.1 | -0.1 | 1.4 | -0.1 | 1.0 | -0.1 | 1.2 |
| **Habit strength (1 [low] to 7 [high])** | | | | | | | | | | | | | | | | | | |
| Baseline | 4.1 | 1.4 | 4.4 | 1.5 | 4.3 | 1.4 | 4.2 | 1.4 | 4.2 | 1.4 | 4.2 | 1.4 | 4.4 | 1.5 | 4.3 | 1.4 | 4.3 | 1.4 |
| Follow-up | 4.2 | 1.4 | 4.3 | 1.4 | 4.3 | 1.4 | 4.2 | 1.4 | 4.3 | 1.4 | 4.2 | 1.4 | 4.2 | 1.4 | 4.4 | 1.4 | 4.3 | 1.4 |
| Difference | 0.1 | 1.0 | -0.1 | 1.1 | 0.0 | 1.0 | 0.1 | 0.9 | 0.0 | 1.9 | 0.1 | 1.0 | -0.2 | 1.2 | 0.1 | 1.1 | -0.1 | 1.1 |
| **Action control (1 [low] to 7 [high])** | | | | | | | | | | | | | | | | | | |
| Follow-up | 4.5 | 1.3 | 4.5 | 1.4 | 4.5 | 1.3 | 4.6 | 1.3 | 4.7 | 1.3 | 4.6 | 1.3 | 4.4 | 1.4 | 4.7 | 1.4 | 4.6 | 1.4 |
| **Difficulty using the planning tool (1 [not at all hard] to 5 [extremely hard])** | | | | | | | | | | | | | | | | | | |
| Baseline (immediately after intervention) | 1.5 | 0.0 | 1.6 | 0.9 | 1.5 | 0.9 | 1.4 | 0.8 | 1.7 | 1.1 | 1.5 | 0.9 | 1.7 | 1.0 | 1.4 | 0.8 | 1.5 | 0.9 |
| **Preferred action plan at follow-up (N, percent)** | | | | | | | | | | | | | | | | | | |
| Literacy-sensitive | 109 | 50.5 | 96 | 52.7 | 205 | 51.5 | 155 | 61.8 | 72 | 56.3 | 227 | 59.9 | 93 | 49.5 | 105 | 49.5 | 198 | 49.5 |
| Standard | 58 | 26.9 | 46 | 25.3 | 104 | 26.1 | 58 | 23.1 | 26 | 20.3 | 84 | 22.2 | 52 | 27.7 | 64 | 30.2 | 116 | 29.0 |
| Not sure | 49 | 22.7 | 40 | 22.0 | 89 | 22.4 | 38 | 15.1 | 30 | 23.4 | 68 | 17.9 | 43 | 22.9 | 43 | 20.3 | 86 | 21.5 |

(mean = 5.3, SD = 1.3) (Table 2). The average habit strength was near the midpoint of the scale (mean = 4.3, SD = 1.4). Across all arms, the average health literacy score (NVS) was 3.6 (Table C in S2 Text; SD = 2.0). This corresponds to just over half of the sample in each arm scoring in the 'high' health literacy category (Table 1).

Overall, 66.5% ($n$ = 1,177) of the participants completed follow-up. There was no difference in completion rates across arms ($\chi^2[2]$ = 0.99, $p$ = 0.61) (Table D in S2 Text). Participants lost to follow-up were more likely to be female (53.2% of non-completers compared to 61.8% of completers; $\chi^2[1]$ = 11.93, $p$ < 0.001; odds ratio 1.43 [95% CI: 1.17 to 1.74]) and were more likely to report a diagnosis of type 2 diabetes (38.9% of non-completers compared to 24.9% of completers; $\chi^2[1]$ = 36.85, $p$ = 0.001; odds ratio 1.92 [95% CI: 1.55 to 2.37]) (Table D in S2 Text). On average, participants lost to follow-up were also younger (−1.3 years; $F_{(1,1767)}$ = 5.1, $p$ = 0.025; 95% CI: −2.48 to −0.17) and reported stronger intentions to reduce their unhealthy snacking (+0.2; $F_{(1,1767)}$ = 12.3, $p$ < 0.001; 95% CI: 0.1 to 0.4) (Table E in S2 Text). There was no statistical evidence of differences in completion rates by baseline BMI, health literacy, snacking score, or habit strength (Tables D and E in S2 Text).

## Aim 1: Effects of health literacy and action plan

**Primary outcome (snacking score).**   Within the randomised arm, participants using the literacy-sensitive plan reported lower baseline habit strength compared to participants using the standard action plan (Table 2). Given that habit strength is an important theoretical moderator of the impact of this intervention [45,46] and the small difference across action plan groups, this variable was included as a covariate in the regression models. This covariate was not explicitly included in the analysis plan outlined in the registered trial information.

Controlling for the effects of age, language spoken at home, education, diabetes status, baseline habit strength, and baseline snacking score, a person who scored 1 standard deviation (2.0 points) below the mean NVS score (lower health literacy) would be predicted to consume 10.0 fewer serves of snacks per week using the literacy-sensitive action plan rather than the standard action plan (95% CI: 0.05 to 19.5; $p$ = 0.039). By contrast, a person who scored 1 standard deviation above the mean NVS score (higher health literacy) would be predicted to consume 3.0 fewer serves of snacks per week using the standard action plan, rather than the literacy-sensitive action plan (95% CI: −6.3 to 12.2; $p$ = 0.529) (Fig 2A); however, this reduction did not reach statistical significance. In addition, we did not observe a statistically significant action plan × health literacy (NVS) interaction ($b$ = −3.25; 95% CI: −6.55 to 0.05; $p$ = 0.054) (Table 3; Table F in S2 Text). Estimates from the intention-to-treat best- and worst-case scenarios were consistent with those from a complete case analysis (Tables G and H in S2 Text). As shown in Fig 2, the relative benefit of the literacy-sensitive action plan compared to the standard action plan reversed as NVS scores moved beyond 5, though note the confidence intervals are overlapping.

Although no participants reported the maximum possible absolute change in snacking score (630 serves) that could be obtained using the instrument, a small number of plausible but extreme values were observed (differences of up to 430.5 serves of unhealthy snacks per week across time points). Regression model coefficients after removal of 8 influential and outlying observations are shown in Table 3 and visualised in Fig 2B. Whilst the model coefficients are smaller in size, the direction of their effects are consistent with the initial analysis, and removal of these observations does not importantly change interpretation of the model. In this model, there was evidence of an interaction between health literacy and action plan ($b$ = −2.07; 95% CI: −3.89 to −0.26; $p$ = 0.025).

**Table 3. Multiple linear regression model predicting reduction in serves of unhealthy snacks per week (randomised arm) [a].**

| Predictor | Unadjusted | | Adjusted[b] | |
|---|---|---|---|---|
| | b (95% CI) | p-Value | b (95% CI) | p-Value |
| **All observations** | | | | |
| Health literacy (NVS) | 3.88 (1.30, 6.47) | 0.003 | 4.21 (1.73, 6.69) | <0.001 |
| Action plan = literacy-sensitive action plan[c] | 0.77 (−6.12, 7.65) | 0.827 | 3.53 (−3.12, 10.18) | 0.299 |
| Action plan = literacy-sensitive action plan × health literacy (NVS) | −3.20 (−6.66, 0.25) | 0.069 | −3.25 (−6.55, 0.05) | 0.054 |
| **Outlier and influential observations removed** | | | | |
| Health literacy (NVS) | 1.98 (0.60, 3.37) | <0.001 | 1.99 (0.64, 3.34) | <0.001 |
| Action plan = literacy-sensitive action plan[c] | 1.57 (−2.11, 5.25) | 0.403 | 2.44 (−1.17, 6.04) | 0.186 |
| Action plan = literacy-sensitive action plan × health literacy (NVS) | −2.11 (−3.98, −0.24) | 0.027 | −2.07 (−3.89, −0.26) | 0.025 |

[a]Analysis uses only participants allocated to the arm 'random (prior preference not assessed)'.

[b]Adjusted analysis controls for mean-centred baseline snacking score, age, English spoken at home, education, self-reported diabetes status, and baseline habit strength.

[c]Action plans were coded such that 1 = literacy-sensitive, 0 = standard.

NVS, Newest Vital Sign.

There was also evidence of an effect of diabetes status ($b = −10.03$; 95% CI: −17.79 to −2.27; $p = 0.001$), indicating that participants with type 2 diabetes were estimated to consume 10.03 more serves of unhealthy snacks per week at follow-up compared to those without diabetes (Table F in S2 Text). However, this effect was not robust to removal of outlier and influential observations ($b = −0.82$; 95% CI: −5.07 to −3.43; $p = 0.706$) (Table I in S2 Text).

**Secondary outcomes.** On average, participants in the randomised arm reported scores for perceived extent of unhealthy snacking that were 0.4 points higher on the scale (perceptions of greater extent of unhealthy snacking) at follow-up than at baseline (SD = 3.22). Health literacy, action plan, and diabetes status did not have significant effects on perceived extent of unhealthy snacking, controlling for the effects of age, language spoken at home, education, diabetes status, baseline habit strength, and baseline perceived extent of unhealthy snacking (Table P in S2 Text).

On average, participants reported that the action plans were easy to use (mean = 1.5, SD = 0.9), and 66.5% ($n = 391$) reported that they were 'not at all hard' to use. More people with high health literacy rated the tool 'not at all hard' to use (78.5%, $n = 267$) than people with low/moderate health literacy (50.0%, $n = 124$; $\chi^2[2] = 66.40$, $p < 0.001$). There was no observed difference in difficulty between the 2 types of action plans ($\chi^2[2] = 0.89$, $p = 0.64$).

**Action plan characteristics.** For literacy-sensitive action plans in the randomised arm, the most frequently selected scenario was 'I have a craving' ($n = 56$, 18.5%) and the most frequently selected plan was 'eat a piece of fruit' ($n = 87$, 28.8%) (Table J in S2 Text). Sixty-eight participants (22.5%) were advised to choose an easier plan because they had rated their plan greater than or equal to 7 on the 10-point scale of plan difficulty. Of these, 54 participants (79.4%) revised their plan.

In the randomised arm, participants with low/moderate health literacy created plans that had an average of 13.6 words (SD = 12.7). Participants with high health literacy created plans having an average of 23.0 words (SD = 20.7). Participants with high health literacy more frequently identified a situation that made them more likely to snack on unhealthy foods (76.2%) compared to people with low/moderate health literacy (42.1%) (Table K in S2 Text). This

group also more frequently identified a plan to reduce unhealthy snacking (90.1% versus 57.9%, respectively), and more often personalised their situations and plans beyond the options available in the literacy-sensitive action plan (Table K in S2 Text).

### Aim 2: Effects of allocation method

**Primary outcome (snacking score).**   Approximately half of the participants in the randomised arm were assigned to use the literacy-sensitive action plan ($n$ = 302, 51.4%). For those in the choice arm, this proportion was 65.7% ($n$ = 383), whereas this was 44.6% in the screened arm ($n$ = 267) (Table 1). In the choice arm, participant health literacy scores were similar across action plans, suggesting participants were not necessarily choosing an action plan that matched their health literacy (mean difference = 0.07; 95% CI: −0.26 to 0.39; $t_{(581)}$ = 0.389, $p$ = 0.70) (Table C in S2 Text).

There was no overall effect of allocation method on reduction in unhealthy snacking, controlling for the effects of health literacy, action plan, age, language spoken at home, education, diabetes status, baseline habit strength, and baseline snacking score ($\chi^2[2]$ = 3.27, $p$ = 0.19). On average, participants in the screened allocation method group reported a reduction of 1.79 serves of unhealthy snacks per week compared to those in the choice allocation method group, though this difference did not reach statistical significance and 95% confidence intervals suggest a true effect within the range of −0.16 to 3.73 ($p$ = 0.067) (Table 4; Table L in S2 Text). There was an effect of health literacy (as measured by NVS) such that each 1-point increase in NVS score (indicating higher health literacy) estimated that the individual would consume 1.47 fewer serves of unhealthy snacks per week at follow-up (95% CI: 0.61 to 2.32; $p$ = 0.001).

There was no evidence of an effect of action plan on snacking score ($b$ = 0.79; 95% CI: −2.61 to 4.19; $p$ = 0.633) (Table L in S2 Text). People who reported having type 2 diabetes were estimated to consume 6.07 more serves of unhealthy snacks at follow-up compared to those without type 2 diabetes (95% CI: −9.79 to −2.35; $p$ = 0.001). Point estimates for best- and worst-case scenarios were consistent with those for completed cases (Tables M and N in S2 Text).

Twenty-four influential observations were sequentially removed from the model individually. The resulting regression model coefficients are consistent with the initial analysis ('All observations', Table 4) but indicate a much smaller mean difference between participants in the screened and choice arms that did not reach statistical significance ($b$ = 0.45; 95% CI: −0.71 to 1.59; $p$ = 0.432) (Table 4; Table O in S2 Text). The main effects of allocation method ($\chi^2[2]$ = 1.21, $p$ = 0.54), health literacy ($b$ = 0.16; 95% CI: −0.35 to 0.67; $p$ = 0.553), and diabetes status ($b$ = −0.32; 95% CI: −2.54 to 1.90; $p$ = 0.763) were not robust to removal of outlier and influential observations.

**Secondary outcomes.**   There were no main effects of allocation method or diabetes status on perceived extent of unhealthy snacking, habit strength, or action control (Tables Q–S in S2 Text). There was no difference in perceived difficulty using the action plan across allocation methods ($\chi^2[4]$ = 6.39, $p$ = 0.17).

## Discussion

This study found that, in a sample of people with overweight/obese BMI or diabetes, the effectiveness of an action plan to reduce unhealthy snacking may apparently depend on the individual's health literacy level; however, these findings did not reach statistical significance. The direction of this relationship is consistent with a novel effect that has received little attention in the literature, and for a group for whom reducing unhealthy snacking is of clinical benefit. On average, people with lower health literacy reported less unhealthy snacking using an action

**Table 4. Multiple linear regression model predicting reduction in serves of unhealthy snacks per week, by allocation arm.**

| Predictor | Unadjusted | | Adjusted[a] | |
|---|---|---|---|---|
| | *b* (95% CI) | *p*-Value | *b* (95% CI) | *p*-Value |
| **All observations** | | | | |
| Health literacy (NVS) | 1.28 (0.47, 2.10) | 0.002 | 1.47 (0.61, 2.32) | 0.001 |
| Allocation method contrast 1: Randomised versus choice/screened | 0.15 (−2.11, 2.40) | 0.908 | 0.16 (−2.04, 2.37) | 0.898 |
| Allocation method contrast 2: Choice versus screened | 1.89 (−0.07, 3.86) | 0.057 | 1.79 (−0.16, 3.73) | 0.067 |
| **Outlier and influential observations removed** | | | | |
| Health literacy (NVS) | 0.15 (−0.34, 0.63) | 0.592 | 0.16 (−0.35, 0.67) | 0.553 |
| Allocation method contrast 1: Randomised versus choice/screened | 0.62 (−0.72, 1.95) | 0.374 | 0.55 (−0.76, 1.85) | 0.421 |
| Allocation method contrast 2: Choice versus screened | 0.69 (−0.47, 1.85) | 0.225 | 0.45 (−0.71, 1.59) | 0.432 |

[a]Adjusted analysis controls for mean-centred baseline snacking score, age, English spoken at home, education, self-reported diabetes status, and baseline habit strength.

NVS, Newest Vital Sign.

plan that employed health literacy principles, whereas people with higher health literacy appeared to benefit more from using a 'standard' version of the action plan. Although the interaction effect did not reach statistical significance in the full sample, this effect was significant in our sensitivity analysis. Results also suggested that the intervention overall was less effective for people with diabetes, and that screening for health literacy could be a more effective method of allocating an action plan than allowing the participant to select their preferred plan; however, the effect estimate for the latter observation did not reach statistical significance. These last 2 observations should be interpreted with caution as they were influenced by a small number of participants with extreme (yet plausible) values.

This study raises questions about health literacy theory and intervention implementation. On the one hand, tailoring an action plan to a person's health literacy level may improve plan effectiveness. If this is the case, this directly challenges the assumptions of a universal precautions approach to health literacy [7] (i.e., that everyone benefits from simplified health materials), at least as it applies to a behaviour change action plan intervention. On the other hand, tailoring appeared to provide a substantial benefit for people with low/moderate health literacy but only a comparatively modest benefit for those with high health literacy. However, these findings were not definitive: The interaction effect, which would support the argument for tailoring, approached but did not achieve statistical significance in the full sample but was significant in our sensitivity analysis. A universal precautions approach (i.e., only providing the literacy-sensitive action plan) may be more appropriate in this context than tailoring, given the aim of reducing health inequality. Further, in terms of implementation, the effect of screening for health literacy approached but did not achieve statistical significance in the full sample and was not significant in our sensitivity analysis. A practical compromise may be to present the literacy-sensitive plan as the 'default' option.

This study has several strengths. The direction of findings was consistent with our previous findings [23] in a more applied sample for whom reduced snacking would be of clinical benefit. This study also used a validated instrument that has high relevance to the Australian context as it maps directly to national guidelines for healthy eating [39]. Furthermore,

overconsumption of the unhealthy snacks targeted in this study has been identified as a key challenge to diet quality in Australia [39]. In addition, the snacking plans are a low-cost and easy-to-use tool that could be feasibly incorporated into existing clinical practice.

We also acknowledge this study's limitations. We observed low–moderate retention (67%). This was unexpected given high retention in the previous study (85%) [23]. There was no difference in retention for people with low, moderate, and high health literacy, but people with diabetes were less likely to complete follow-up, and this should be addressed in future research. Further, low–moderate retention reduced the number of participants in each arm available for per protocol analysis and may have reduced power to detect statistically significant effects.

Another limitation is that the action plan preferences of participants in the choice arm were informed by the names—Smart Snacking 101 (basic) and Smart Snacking Pro (advanced)—and descriptions of our interventions. Whilst we intended these to accurately contrast the 2 action plans and present both in a positive light, these descriptions may have generated expectations and potentially influenced intervention appraisal and effectiveness. We also acknowledge the inherent limitation of a once-off measure of snacking behaviour. This approach reduces participant burden and minimises the risk of missing data, but the resulting estimates may be less accurate than those for electronic diaries. Lastly, we acknowledge that the inclusion of participants with extreme values for self-reported unhealthy snacking renders interpretation more difficult. Given that these responses cannot be ruled out as impossible, we have provided a sensitivity analysis to maintain transparency of these findings.

Future studies should continue to investigate the relationship between a person's health literacy and behaviour change intervention effectiveness. This could include, for example, investigating this relationship for other health outcomes and other 'literacy-sensitive' variants of behaviour change strategies. For example, upon reflection, the measure of unhealthy snacking in this study adhered to health literacy principles and could be evaluated as a behaviour change strategy in itself (self-monitoring). As additional literacy-sensitive variants are developed, the value of tailoring to health literacy versus a universal precautions approach should continue to be evaluated (on a case-by-case basis). Additional research is also required to establish whether the effects observed in this study are sustained in the long term, including whether effects attenuate differently for people with low and high health literacy. Given that low health literacy is more prevalent amongst people who speak English as a second language and who were born overseas [2,5], this program of research could also benefit from developing culturally adapted versions of the snacking tool. This is particularly important given the influence of culture on diet [47,48].

It is unknown what impact assessment of participant preferences might have on action plan effectiveness. The findings from this study highlight the potential limitations of conventional preference trial analyses. Further (a priori) analysis is warranted to estimate the independent contributions of a person receiving their preferred action plan and the effects of self-selection. As outlined in our previously published protocol [35], this research question will be addressed by analysis of an additional arm not described in this paper (Arm A1 in Fig 1). Analysis pertaining to this research question will be published elsewhere.

It is also worth reflecting on the strong influence of a small number of participants (2% of the sample that completed follow-up) on reduced unhealthy snacking, particularly with regards to the effect of diabetes status. Although consumers with type 2 diabetes were consulted for feedback on the intervention instructions and the appropriateness of the literacy-sensitive action plans [35], future studies could seek involvement from additional individuals with type 2 diabetes who explicitly identify themselves as consuming a large number of unhealthy snacks (e.g., more than 100 serves per week).

In conclusion, this study found an apparent direction of effects suggesting that for people with overweight/obese BMI or diabetes, action plans to reduce unhealthy snacking may be more effective for people with lower health literacy when they employ health literacy principles. In contrast, 'standard' action plans that offer greater flexibility to personalise plans may be more effective for people with higher health literacy. However, these effects should be interpreted with caution as they did not reach statistical significance. This study highlights the importance of investigating differential effects of health literacy on behavioural interventions, which has received little attention in the literature. In addition, we did not observe significant differences in effects on snacking when plans were allocated through health literacy screening or allowing the individual to choose. Given the lack of a clear benefit of either allocation method, and the absence of a statistically robust benefit of the standard action plan for people with higher health literacy, it may be more practical (and more cost-effective) to present the literacy-sensitive action plan as the 'default choice' rather than tailor to health literacy.

## Supporting information

**S1 Data.**
(XLSX)

**S1 Surveys. Baseline and follow-up surveys.**
(PDF)

**S1 Text. Protocol and CONSORT and TIDieR checklists.**
(PDF)

**S2 Text. Additional tables.**
(PDF)

## Acknowledgments

We would like to thank our type 2 diabetes consumer health representatives Edward Hartley and Mike Font for their feedback and input into the design of this study. We would also like to acknowledge the CSIRO for permission to use items from the discretionary foods components of the CSIRO diet score.

## Author Contributions

**Conceptualization:** Julie Ayre, Erin Cvejic, Carissa Bonner, Robin M. Turner, Stephen D. Walter, Kirsten J. McCaffery.

**Data curation:** Julie Ayre, Kirsten J. McCaffery.

**Formal analysis:** Julie Ayre, Erin Cvejic, Carissa Bonner, Robin M. Turner, Stephen D. Walter, Kirsten J. McCaffery.

**Funding acquisition:** Julie Ayre, Carissa Bonner, Robin M. Turner, Kirsten J. McCaffery.

**Investigation:** Julie Ayre.

**Methodology:** Julie Ayre, Erin Cvejic, Carissa Bonner, Robin M. Turner, Stephen D. Walter, Kirsten J. McCaffery.

**Project administration:** Julie Ayre, Carissa Bonner.

**Resources:** Julie Ayre.

**Software:** Julie Ayre, Erin Cvejic.

**Supervision:** Erin Cvejic, Carissa Bonner, Kirsten J. McCaffery.

**Validation:** Erin Cvejic.

**Visualization:** Julie Ayre.

**Writing – original draft:** Julie Ayre.

**Writing – review & editing:** Julie Ayre, Erin Cvejic, Carissa Bonner, Robin M. Turner, Stephen D. Walter, Kirsten J. McCaffery.

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
