## [Editor Report · Decision Letter 0]

6 Apr 2020

Dear Dr Ayre, 

Thank you for submitting your manuscript entitled "Effects of health literacy, screening and participant choice on action plans for reducing unhealthy snacking: A randomised controlled trial" for consideration by PLOS Medicine.

Your manuscript has now been evaluated by the PLOS Medicine editorial staff and I am writing to let you know that we would like to send your submission out for external peer review.

Kind regards,

Helen Howard, for Clare Stone PhD 

Acting Editor-in-Chief

PLOS Medicine 

plosmedicine.org

---

## [Decision Letter · Decision Letter 1]

13 May 2020

Dear Dr. Ayre,

Thank you very much for submitting your manuscript "Effects of health literacy, screening and participant choice on action plans for reducing unhealthy snacking: A randomised controlled trial" (PMEDICINE-D-20-00842R1) for consideration at PLOS Medicine. 

[LINK]

In light of these reviews, I am afraid that we will not be able to accept the manuscript for publication in the journal in its current form, but we would like to consider a revised version that addresses the reviewers' and editors' comments. Obviously we cannot make any decision about publication until we have seen the revised manuscript and your response, and we plan to seek re-review by one or more of the reviewers. 

We expect to receive your revised manuscript by Jun 03 2020 11:59PM. Please email us (plosmedicine@plos.org) if you have any questions or concerns.

We look forward to receiving your revised manuscript. 

Sincerely,

Emma Veitch, PhD

PLOS Medicine

On behalf of Clare Stone, PhD, Acting Chief Editor,

PLOS Medicine

plosmedicine.org

*In the last sentence of the Abstract Methods and Findings section, please include a brief description of any key limitation(s) of the study/study methods.

Comments from the reviewers:

Reviewer #1: See attachment

Michael Dewey

Reviewer #2: The purpose of the study was to evaluate: 1) the impact of health literacy and action plan type on unhealthy snacking, for people who have type 2 diabetes or high BMI; and 2) the most

effective method of action plan allocation. This is an interesting study but needs to be enhanced in terms of readability and other issues below.

1. The research problem, study design, analysis, and the results need to be stated in a consistent way. 

Hypothesis 1: The authors stated that there is a need to identify the effect of tailored action plan for people with low and high health literacy (p.5), but a recent study already showed the findings in the introduction (p.6). Then, what is the value of this study (hypothesis 1), in addition to the previous findings? 

Hypothesis 2: What is action plan allocation in terms of practical usefulness? Did the authors want to compare the effect of tailored action plan after HL screening with the patients' choice of preferred action plan? If so, subgroup comparison between the group received tailored action plan and the group preferred action plan would be sufficient. It seems that there is no need to compare these groups with random allocation group.

2. There might be no need to address about research questions not analyzed in this study (p.7, 3rd paragraph; p.8, line 5-7).

3. Please describe the study design in detail. 

Please describe about reliability and validity of the measurement.

4. Please report the results according to the research questions.

Reviewer #3: See attachment

[LINK]

---

## [Decision Letter · Decision Letter 2]

30 Jun 2020

Dear Dr. Ayre,

Thank you very much for submitting your manuscript "Effects of health literacy, screening and participant choice on action plans for reducing unhealthy snacking: A randomised controlled trial" (PMEDICINE-D-20-00842R2) for consideration at PLOS Medicine. 

Your paper was evaluated by a senior editor and discussed among all the editors here. It was also discussed with an academic editor with relevant expertise, and sent to a statistical reviewer. The reviews are appended at the bottom of this email and any accompanying reviewer attachments can be seen via the link below:

[LINK]

In light of these reviews, I am afraid that we will not be able to accept the manuscript for publication in the journal in its current form, but we would like to consider a revised version that addresses the reviewers' and editors' comments. Obviously we cannot make any decision about publication until we have seen the revised manuscript and your response, and we plan to seek re-review by one or more of the reviewers. 

We expect to receive your revised manuscript by Jul 21 2020 11:59PM. Please email us (plosmedicine@plos.org) if you have any questions or concerns.

We look forward to receiving your revised manuscript. 

Sincerely,

Clare Stone, PhD

Managing Editor 

PLOS Medicine

plosmedicine.org

In general, the presentation needs to be significantly improved for publication in a medical journal and to more closely follow reporting for a clinical trial. For example, in the abstract no geographic, demographic, date information, intervention or outcomes are provided. These are all needed for trial publications. I have included more specifics below. Please do look at reporting on other trials for guidance. I am afraid the presentation issues are significant and should be resolved before we are able to proceed with your manuscript. 

When resubmitting please resubmit with line numbers.

Title add a country setting, please.

In the abstract you mention the ‘universal precautions approach’ – firstly can you please state what this is, briefly (who signs up to it etc) and also please remove quote marks.

Abstract – please avoid a listy type of prose “Two-stage randomised controlled trial.” instead, we performed….

Abstract – and elsewhere, please be clear on what is statistically significant (with p values provided) and what’s not.

Abstract Methods and Findings: * Please ensure that all numbers presented in the abstract are present and identical to numbers presented in the main manuscript text. * Please include the study design, population and setting, number of participants, years during which the study took place, length of follow up, and main outcome measures (note you don’t list the outcomes as yet) * Please quantify the main results (with 95% CIs and p values). * Please include the important dependent variables that are adjusted for in the analyses. * Please include the actual amounts and/or absolute risk(s) of relevant outcomes (including NNT or NNH where appropriate), not just relative risks or correlation coefficients. (example for absolute risks: PMID: 28399126). * Please include a summary of adverse events if these were assessed in the study. Please ensure summary demographic information is added;* In the last sentence of the Abstract Methods and Findings section, please describe the main limitation(s) of the study's methodology.

Data - PLOS does not permit "data not shown or data is available on request (from the authors).” Please remove this claim, or do one of the following: a) If you are the owner of the data relevant to this claim, please provide the data in accordance with the PLOS data policy, and update your Data Availability Statement as needed. b) If the data not shown refer to a study from another group that has not been published, please cite personal communication in your manuscript text (it should not be included in the reference section). Please provide the name of the individual, the affiliation, and date of communication. The individual must provide PLOS Medicine written permission to be named for this purpose. c) For any other circumstance, please contact me ASAP.

“chunking information” – please amend to perhaps, breaking into smaller / manageable…

“The current paper” – the current ‘study’

Again, on presentation please do read published trials to see presentation – in the introduction we need more of the core information presented, interventions, outcomes and so on

Page 9 – you say “Participants were excluded if they did not speak English, but they were not excluded on the basis of their snacking behaviour.” And in Fig 1 you say exclusions were based on BMI. Im confused – can you please clarify / be consistent?

How is health literacy assessed / measured? 

Please ensure all questionnaires are provided as Supp files. 

Snacking scores section – please be explicit – what is the primary outcome? I think you have used the header to tell us but we need it in sentence form. 

Secondary outcomes – again it is written in a listy way. Instead of writing the outcome as the header, with resulting multiple headers, please state in a sentence what the secondary outcomes are, how they’re measured and what the follow up time is (and these should match your trial registration page). There should be one header “Secondary outcomes”

All quantifiable data needs to have accompanying 95%Cis and p values and state is significant.

Results section – please go through this and compare to Table 1 – I find it very confusing. 

For example you say “1,769 participants at baseline, of whom 523….” And “Overall, 66.5% (N=1,177) of the participants completed follow-up” perhaps im confused, but this doesn’t seem to match Figure 1. Also page 17 you say “more likely to report a diagnosis of type 2 diabetes” yet in Figure 1 those with diabetes are excluded and you also refer on pages 25 and 26 and elsewhere to those with diabetes.

Please ensure that numbers in all tables match those given in Fig 1. 

CONSORT checklist – please use sections and paragraphs as page numbers change during revisions, etc.

Did your study have a prospective protocol or analysis plan? Please state this (either way) early in the Methods section. a) If a prospective analysis plan (from your funding proposal, IRB or other ethics committee submission, study protocol, or other planning document written before analyzing the data) was used in designing the study, please include the relevant prospectively written document with your revised manuscript as a Supporting Information file to be published alongside your study, and cite it in the Methods section. A legend for this file should be included at the end of your manuscript. b) If no such document exists, please make sure that the Methods section transparently describes when analyses were planned, and when/why any data-driven changes to analyses took place. c) In either case, changes in the analysis-- including those made in response to peer review comments-- should be identified as such in the Methods section of the paper, with rationale.

Comments from the reviewers:

Reviewer #1: The authors have addressed my points.

As I originally mentioned the question of what to do about possible outliers is a complex one and we do still disagree about possible ways forward. Given the difficulties outlined in the rebuttal I would not want to push this but I would suggest that a sentence be added to the limitations stating that these people are a problem for interpretation.

Michael Dewey

[LINK]

---

## [Editor Report · Decision Letter 3]

18 Aug 2020

Dear Dr. Ayre,

Thank you very much for submitting your manuscript "Effects of health literacy, screening and participant choice on action plans for reducing unhealthy snacking in Australia: A randomised controlled trial" (PMEDICINE-D-20-00842R3) for consideration at PLOS Medicine. 

[LINK]

In light of these reviews, I am afraid that we will not be able to accept the manuscript for publication in the journal in its current form, but we would like to consider a revised version that addresses the reviewers' and editors' comments. Obviously we cannot make any decision about publication until we have seen the revised manuscript and your response, and we plan to seek re-review by one or more of the reviewers. 

We expect to receive your revised manuscript by Aug 25 2020 11:59PM. Please email us (plosmedicine@plos.org) if you have any questions or concerns.

We look forward to receiving your revised manuscript. 

Sincerely,

Clare Stone, PhD

Managing Editor 

PLOS Medicine

plosmedicine.org

It seems issues persist around presentation. Repeatedly in the manuscript, there are claims of "more effective" and "greater reduction" and the like, but it may well be that there are in fact no statistically robust findings. Can you please respond to this and clarify in the manuscript on each occasion. Note, we will not continue with this manuscript if these issues persist as we have requested on more than one occasion. 

Please trim the background subsection of the abstract

Please rephrase "main outcome" as "primary outcome" in the abstract. 

We suggest adapting the abstract to present the findings for "aim 1" and "aim 2" sequentially, reporting the number of participants and primary quantitative observations for each aim. 

Please rename figure 1 "Participant flowchart" or similar. 

Around line 387, you describe "weak evidence" of a difference between groups in unhealthy snacking; these data appear to be those associated with a claim of "more effective" in the abstract, around line 45. Please ensure that the main findings are described in a consistent way - generally, rather than "weak evidence" we favour "non-significant reduction" or similar language. 

Please report exact p values or p<0.001 (noting the example at line 481). 

In the attached CONSORT and Tidier checklists, please refer to individual items by section and paragraph numbers rather than by line or page numbers. 

Comments from the reviewers:

[LINK]

---

## [Editor Report · Decision Letter 4]

14 Sep 2020

Dear Dr. Ayre,

Thank you very much for re-submitting your manuscript "Effects of health literacy, screening and participant choice on action plans for reducing unhealthy snacking in Australia: A randomised controlled trial" (PMEDICINE-D-20-00842R4) for review by PLOS Medicine.

I have discussed the paper with my colleagues and the academic editor. The remaining editorial and production issues need to be fully resolved, including the use of appropriate language to distinguish the significant from the non-significant results, before we can accept the paper for publication in the journal.

[LINK]

We look forward to receiving the revised manuscript by Sep 21 2020 11:59PM. 

Sincerely,

Caitlin Moyer, PhD

Associate Editor

PLOS Medicine

on behalf of

Clare Stone, PhD

Managing Editor 

PLOS Medicine

plosmedicine.org

Requests from Editors:

1. Response to reviewer/editor comments: Thank you for your response that “Our revisions seek to place greater emphasis on estimates of the effects and their associated confidence intervals. We have also softened language around ‘more effective’ and ‘greater reduction’ where confidence intervals indicated that no effect may be present. We feel this approach better aligns with the 2016 ASA statement on p-values (https://doi.org/10.1080/00031305.2016.1154108) that suggests scientific conclusions should not be based only on whether a p-value passes a specific threshold. As in the previous version, we continue to report p values and now report exact p values, as requested. We hope this provides a more nuanced and precise description of the data.”

Please note that PLOS Medicine policy requires that findings be presented with both 95% CIs and p values. Further, in instances where confidence intervals (and p values) indicate that no effect is present, the language needs to convey that no significant effect was found. For example, in the author summary, “The literacy-sensitive action plan may be more effective for participants with low moderate health literacy, whereas the standard action plan may be more effective for those with high health literacy” is misleading as you did not observe a significant interaction effect. Please revise similar instances repeated throughout the manuscript.

2. Abstract: Methods and Findings: Line 33-36: Please also summarize the secondary outcome measures, and if space permits please present the findings (though we understand that word limits of abstract may preclude this).

3. Abstract: Lines 39-44: For this sentence, please include p values in addition to the 95% CIs presented for the fewer servings per week consumed by those with 1SD lower than average health literacy using the literacy sensitive plan (please also specify that this is compared with the standard plan). 

4. Please include the p values for the fewer servings consumed by those 1SD above the mean health literacy score on the standard plan (and please also mention that this is in comparison to the literacy sensitive plan).

In addition to adding the p values, please do not use italics for emphasis, and we suggest revising to: “... however, we observed a non-significant no action plan × health literacy (NVS) interaction (b= -3.25, 95% CI: -6.55 to 0.05, p=0.054).” or similar to highlight that this interaction did not reach statistical significance.

5. Abstract: Line 45-46: “It is unclear which allocation method is more effective.” Please clarify what is meant by this sentence, e.g. by presenting your results.

6. Abstract: "Tailoring to health literacy may improve effectiveness of action plans, but costs of this strategy compared to universal precautions approach needs further investigation.”

This sentence does not appear to accurately summarize what can be concluded from your study. We suggest: “In our study we observed nominal improvements in effectiveness of action plans tailored to health literacy, however these improvements did not reach statistical significance, and the costs associated with such strategies compared with universal precautions needs further investigation.”

7. Author Summary: Why was this study done?: Please do not use italics for emphasis.

8. Author Summary: What did the researchers do and find? Please revise the following points to accurately reflect your study (or similar): 

--The literacy-sensitive action plan may be more effective for participants with low 

moderate health literacy, whereas the standard action plan may be more effective for those with high health literacy. However, these effects did not reach statistical significance. 

-- We did not find evidence suggesting that allocation had a significant effect on reported snacking behavior

9. Author summary: What do these findings mean?: Please revise the second bullet point to “However, this benefit did not reach statistical significance and should be weighed against the cost of tailoring compared to providing everyone with the same (literacy-sensitive) intervention (universal precautions approach).”

10. Introduction: Please consider shortening the Introduction section, particularly the information presented on pages 6-8. For example: 

Lines 115-136: Please reduce the description of your previous study in the Introduction and summarize the findings more succinctly. 

Lines 141-147: The information in this paragraph could be made more succinct, or presented elsewhere.

11. Introduction: Line 160-167: When presenting your question or hypothesis, please use paragraph format rather than a numbered list.

12. Methods: Line 176: Prospective plan- Please include the study protocol document and analysis plan, with any amendments, as Supporting Information to be published with the manuscript if accepted.

13. Results: Lines 383- 390: Please provide the p values to accompany the 95% CIs for the following results: “Controlling for the effects of age, language spoken at home, education, habit strength, and baseline snacking scores, a person who scored one standard deviation (2.0 points) below the mean NVS score (lower health literacy) would be predicted to consume 10.0 fewer serves of snacks per week using the literacy-sensitive action plan rather than the standard action plan (95% CI: 0.05 to 19.5). By contrast, a person who scored one standard deviation above the mean NVS score (higher health literacy) would be predicted to consume 3.0 fewer serves of snacks per week using the standard action plan, rather than the literacy-sensitive action plan (95% CI: -6.3 to 12.2) (Fig 2 Panel A). 

14. Results: Lines 390-392: Please rephrase this sentence to read: “However, we did not observe a statistically significant action plan × health literacy (NVS) interaction (b= -3.25, 95% CI: -6.55 to 0.05,p=0.054) (Table 3 and S3 Table S6).

15. Results: Lines 393-395: Please revise this sentence as follows to note that confidence intervals are overlapping/there is no statistically significant reversal. “ As shown in Fig 2, the relative benefit of the literacy-sensitive action plan compared to the standard action plan is apparently reversed as NVS scores moved beyond 5; however, the confidence intervals are overlapping” or similar.

16. Results: Please revise to include the result: “However, this effect was not robust to removal of outlier and influential observations (b=-0.82, 95%CI= (-5.07-3.43); p= 0.706) (S3 Table S9).

17.Results: Line 437: Please indicate that “M” stands for “mean” at the first use in the text

18. Results: Lines 469-473: Please revise to “On average, participants in the screened allocation method reported a reduction of 1.79 serves of unhealthy snacks per week compared to the choice allocation method, though this did not reach statistical significance and 95% confidence intervals suggest a true effect within the range of -0.16 to 3.73 (p=0.067) (Table 4 and S3 Table S12).”

19.Results: Lines 488-490: Please revise to: “The resulting regression model coefficients are consistent with the initial analysis (‘All observations’, Table 4), but indicate a much smaller mean difference between participants in the screened and choice arms and did not reach statistical significance (Table 4 and S3 Table S15).” Please also provide results, 95% CIs and p values for the findings for the screened vs. choice comparison, as well as the finding for the main health literacty effect, for the analysis with outliers removed as these were highlighted for the “all data” analysis presented above.

20. Results: Lines 494-495: Please present the results in the text (with 95% CIs and p values) or provide a note to which table contains these findings. “There were no main effects of allocation method or diabetes status on perceived extent of unhealthy snacking, habit strength or action control.”

21.Discussion: Please revise throughout to reflect the study’s findings. Specifically, your results and statistical analyses do not support these conclusions. “This study found that the effectiveness of an action plan to reduce unhealthy snacking may apparently depend on the individual’s health literacy level; however, these findings did not reach statistical significance. The direction of this relationship is consistent with a novel effect that has received little attention in the literature, in a sample of people with diabetes or overweight/obese BMI, for whom reducing unhealthy snacking is of clinical benefit. On average, people with lower health literacy reported less unhealthy snacking using an action plan that employed health literacy principles, whereas people with higher health literacy appeared to benefit more from using a ‘standard’ version of the action plan. Although the interaction effect did not reach statistical significance in the full sample, this effect was significant in our sensitivity analysis. Results also suggested that the intervention overall was less effective for people with diabetes, and that screening for health literacy could be a more effective method of allocating an action plan than allowing the participant to select their preferred plan; however the effect estimate for the latter observation did not reach statistical significance. These last two observations should be interpreted with caution as they were influenced by a small number of participants with extreme (yet plausible) values.

22.Discussion: Lines 518-520: Please revise this sentence to more accurately reflect the findings, namely that the interaction between choice/screening and health literacy approached but did not achieve statistical significance. “On the other hand, tailoring appeared to provide a substantial benefit for people with low to moderate health literacy but only a comparatively modest benefit for those with high health literacy; however, the interaction between health literacy and using a screening approach approached but did not achieve statistical significance and were not robust to the removal of outliers.” or similar.

23.Discussion: Lines 575-579: As this presumably refers to the arm of your study to be analyzed and published separately, please remove this from the discussion: “... but our exploratory findings suggest that a person’s preference for an action plan may also play a role in action plan effectiveness.” and please revise the following to: “Further (a priori) analysis is warranted to estimate the independent contributions of a person receiving their preferred action plan and the effects of self-selection.”

24.Discussion: Limitations section: Please comment on the limitation that you did not achieve the goal of 500 participants per arm and the impact this may have on detecting statistically significant effects.

25. Discussion: Lines 581-585: Please revise to more accurately reflect the study findings. In particular, results were not presented suggesting effects in those with overweight/obese BMI: “In conclusion, this study found that the apparent directions of effects suggesting that for people with diabetes, action plans to reduce unhealthy snacking may be more effective for people with lower health literacy when they employed health literacy principles. In contrast, ‘standard’ action plans that offered greater flexibility to personalise plans may be more effective for people with higher health literacy. However, these effects should be interpreted with caution as they did not reach statistical significance.”

26.Discussion: Lines 587-592: Please also revise to clarify these sentences: “In addition,we did not observe significant differences in effects on snacking when interventions are allocated through health literacy screening or allowing the individual to choose. Given the lack of a clear benefit of either allocation method, and the absence of a statistically robust benefit of the standard action plan for people with higher health literacy, it may be more practical (and more cost-effective) to present the literacy-sensitive action plan as the ‘default choice’

rather than tailor to health literacy.” Or similar.

27. Data Availability Statement: This section can be removed from the main text of the manuscript. It will be extracted from the “Data Availability” section of the manuscript submission form, so please be sure all information there is entered accurately and completely.

28.Checklists: Thank you for including the CONSORT and TIDieR checklists; please remove references to page numbers as these are subject to change. Please instead only use section and paragraphs to refer to locations in text.

29. Figure 2: In the legend, please describe the differences between the two panels. For Panel A: On the Y axis, the negative signs are missing between 10 and 25 for the values between “no change” and “more snacking”.

30. Table 3: There is an extra common in the adjusted results 95% CIs presented for action plan in the top panel (3.53 (-3.12, 10.18,).

[LINK]

---

## [Editor Report · Decision Letter 5]

2 Oct 2020

Dear Ms Ayre, 

On behalf of my colleagues and the academic editor, Dr. Sanjay Basu, I am delighted to inform you that your manuscript entitled "Effects of health literacy, screening and participant choice on action plans for reducing unhealthy snacking in Australia: A randomised controlled trial" (PMEDICINE-D-20-00842R5) has been accepted for publication in PLOS Medicine. 

PRODUCTION PROCESS

Before publication you will see the copyedited word document (within 5 busines days) and a PDF proof shortly after that. The copyeditor will be in touch shortly before sending you the copyedited Word document. We will make some revisions at copyediting stage to conform to our general style, and for clarification. When you receive this version you should check and revise it very carefully, including figures, tables, references, and supporting information, because corrections at the next stage (proofs) will be strictly limited to (1) errors in author names or affiliations, (2) errors of scientific fact that would cause misunderstandings to readers, and (3) printer's (introduced) errors. Please return the copyedited file within 2 business days in order to ensure timely delivery of the PDF proof. 

If you are likely to be away when either this document or the proof is sent, please ensure we have contact information of a second person, as we will need you to respond quickly at each point. Given the disruptions resulting from the ongoing COVID-19 pandemic, there may be delays in the production process. We apologise in advance for any inconvenience caused and will do our best to minimize impact as far as possible.

PRESS

PROFILE INFORMATION

Thank you again for submitting the manuscript to PLOS Medicine. We look forward to publishing it. 

Best wishes, 

Caitlin Moyer

Senior Editor 

PLOS Medicine

plosmedicine.org